# Safe and Efficient Use of Tocilizumab in Rheumatoid Arthritis Patient on Maintenance Hemodialysis: A Case Report

**DOI:** 10.3390/medicina59091517

**Published:** 2023-08-23

**Authors:** Paula Kilić, Lucija Ikić, Miroslav Mayer, Marinko Artuković, Ksenija Maštrović Radončić, Marina Ikić Matijašević

**Affiliations:** 1Department of Clinical Immunology, Rheumatology, and Pulmonology, University Hospital Sveti Duh, 10000 Zagreb, Croatia; marinko.artukovic@gmail.com; 2Department of Anatomy and Physiology, University of Applied Health Sciences, 10000 Zagreb, Croatia; lucijaikic1995@gmail.com; 3Department of Internal Medicine, School of Medicine, University of Zagreb, 10000 Zagreb, Croatia; miro.mayer@gmail.com; 4Division of Clinical Immunology and Rheumatology, University Hospital Centre Zagreb, 10000 Zagreb, Croatia; 5Department of Physical Medicine and Rehabilitation, University Hospital Sveti Duh, 10000 Zagreb, Croatia; ksenija.mastrovic@gmail.com

**Keywords:** rheumatoid arthritis, inflammatory arthritis, tocilizumab, interleukin-6, end-stage renal disease, hemodialysis, kidney transplantation

## Abstract

*Background*: Rheumatoid arthritis (RA) is a chronic systemic autoimmune and inflammatory disease. Conventional synthetic and biologic disease-modifying antirheumatic drugs (DMARDs), Janus kinase inhibitors, and rituximab are used to treat the disease. There are no recommendations or guidelines for the treatment of patients with both inflammatory arthritis and end-stage renal disease (ESRD), despite the safety and efficacy of the mentioned drugs. The anti-interleukin-6 receptor antibody tocilizumab (TCZ) has not been used as a long-term therapy for hemodialysis (HD) patients with RA, except in a few case reports. *Case Description*: We present the case of a 41-year-old patient with RA and ESRD on maintenance HD due to type 1 diabetes-related complications. Due to high RA disease activity, the patient was not a suitable candidate for a kidney transplant. Because TCZ is used to treat both RA and kidney transplant rejection, therapy with a full dose of TCZ was administered. The patient has achieved sustained clinical remission (for the past four years) with no adverse events reported. *Conclusions*: Herein, we present the safe and effective use of TCZ in an RA patient on HD who is also a candidate for kidney transplant. Consequently, TCZ could be the treatment of choice for RA patients with ESRD who have not achieved disease control (low activity or remission) with conventional synthetic DMARDs. Clinical studies are required to evaluate the efficacy and safety of biologic DMARDs and Janus kinase inhibitors in patients with both inflammatory arthritis and ESRD.

## 1. Introduction

Rheumatoid arthritis (RA) is a chronic autoimmune disease characterized by the accumulation of inflammatory cells in the synovium, causing joint destruction, and high levels of pro- and anti-inflammatory cytokines that regulate local synovial and systemic inflammation, causing intra-articular and extra-articular manifestations. Interleukin (IL)-6 is a pro-inflammatory cytokine that has a very important role in the pathogenesis of RA. New studies have shown that this IL also plays a role in the pathogenesis of diabetes and renal injury in glomerulonephritis and other types of renal disease [1,2,3]. Although renal impairment is a relatively common comorbidity in patients with RA (some studies even demonstrate its prevalence of up to 33.8%), being mostly associated with factors such as age, hypertension, and diabetes rather than RA-related factors, the treatment of RA patients with renal disease (especially with end-stage renal disease (ESRD)) remains a challenge for every rheumatologist and nephrologist. Rheumatoid arthritis is currently treated with drugs that are either contraindicated, are used with strict precautions in a dose adjusted for creatinine clearance, or their use in patients with renal insufficiency is yet to be evaluated in clinical studies [4]. In the cross-sectional study published by Paudyal et al. in 2017, the results showed that the co-occurrence of RA with ESRD was 1.1%. While biologic agents accounted for less than 8% of the medications prescribed for the treatment of RA patients with concomitant ESRD, none of the patients received tocilizumab (TCZ) as a therapy [5]. Tocilizumab is a humanized monoclonal antibody aimed against the receptor of IL-6 indicated in combination with methotrexate (MTX), but it has demonstrated its efficacy as a monotherapy in cases of intolerance to MTX or when treatment with MTX is contraindicated, as in hemodialysis (HD) patients. Tocilizumab has a good safety profile, which has also been observed in relation to other biologic agents, but so far, little is known about its safe use in patients with RA on renal replacement therapy [6]. Here, we present a complex case of a patient suffering from three chronic inflammatory diseases (RA, chronic kidney disease, and type 1 diabetes (T1D)) that are associated with higher plasma levels of IL-6 compared to a healthy population and who, despite ESRD and maintenance HD, was successfully and safely treated with anti-IL-6 therapy (TCZ).

## 2. Case Description

A 41-year-old woman was referred to our outpatient clinic due to tenderness and swelling of her metacarpophalangeal and proximal interphalangeal joints of both hands for the previous 10 months. Her anti-cyclic citrullinated peptide (CCP) antibody and rheumatoid factor returned strongly positive, with elevated inflammatory markers C-reactive protein (CRP) and erythrocyte sedimentation rate (ESR) (Table 1).

We diagnosed our patient with RA with a disease activity Score-28 (DAS28) CRP of 5.91, which indicates high disease activity. It warrants mentioning that three years earlier, an orthopedist had treated the patient with local therapy due to short-term tenderness and swelling of her left knee (ultrasound-proven synovitis and effusion) and left wrist (ultrasound-proven extensor tenosynovitis). According to the available medical documentation, she had continuously elevated ESR and CRP for three years prior to consulting us. The patient had been diagnosed with T1D at the age of 9, followed by the later diagnosis of arterial hypertension, hyperlipoproteinemia, and coronary artery disease (CAD). Since the age of 31, she has been monitored due to diabetic nephropathy and retinopathy. Due to the progressive insufficiency of her renal function five years later, she began maintenance HD three times a week, followed by a regular nephrologist’s examination in the transplant center. Due to ESRD, she had a contraindication to MTX; therefore, therapy with prednisone (15 mg/day) and sulfasalazine (SSZ) was given. After one month, her liver transaminases increased, and the SSZ dose was reduced from 2.0 to 1.5 g/day with a consequential normalization of the enzymes. Approximately 6 months later, after initially responding well to the treatment, the patient’s symptoms worsened, and the inflammatory markers increased (most likely due to prednisone discontinuation). The disease activity score-28 CRP was 5.3, suggesting there was still high disease activity and making her an ineligible candidate for renal transplantation. As per an agreement with our clinical pharmacologist, we did not introduce another conventional synthetic disease-modifying antirheumatic drug (csDMARD), such as leflunomide. The concomitant use of two csDMARDs significantly increases the probability of liver toxicity in general in patients, especially in those who are on HD. Since TCZ is also used off-label as a part of immunosuppressive therapy after kidney transplantation, the treating nephrologist agreed with its administration. The patient was thoroughly informed about all the possible known risks of the drug prior to her treatment. The treatment started after securing her consent. Tocilizumab was given subcutaneously once weekly at a dose of 162 mg (immediately after dialysis) as monotherapy, whereas SSZ was discontinued. On the first outpatient clinic visit after 12 weeks of TCZ therapy, the patient achieved remission with a DAS28 CRP of 1.94. During the 4-year period, we did not record any significant or severe adverse event requiring therapy discontinuation. Moreover, our patient better manages her arterial hypertension and diabetes, as shown by her hemoglobin A1c levels. She also has a stable CAD and normal lipid profile. Unfortunately, to our knowledge, due to the COVID-19 pandemic, the number of organ transplants in our country has been significantly reduced and our patient is still waiting for a kidney transplant.

## 3. Discussion

The use of biologic DMARDs (bDMARDs) in severe renal insufficiency remains understudied, despite their presence in clinical practice for more than twenty years. We can presume that kidney failure can affect the metabolism of bDMARDs, leading to drug accumulation and, consequently, increased toxicity. On the other hand, HD can cause excessive drug removal, reducing its efficacy. The manufacturer does not provide dosage adjustments for creatinine clearance (CrCl) < 30 mL/min and there are no studies approving TCZ use in ESRD patients; however, based on tocilizumab’s molecular weight (148 kDa), it is unlikely to be significantly renally eliminated [7]. Ryman and Meibohm published an article in 2017 describing the pharmacokinetics of available monoclonal antibodies, stating that due to their large size, monoclonal antibodies cannot undergo renal filtration and, consequently, urine elimination, but are instead eliminated by excretion or catabolism [8]. Tocilizumab is an IL-6 inhibitor that has shown good results in the treatment of RA patients, especially those unresponsive to csDMARDs [9,10,11]. Its high efficacy could be explained by the key role of IL-6 in RA. The anti-inflammatory effects of TCZ, which exceed current rheumatologic pathology, have been examined in recent years, with a focus on its off-label use in renal transplant patients and COVID-19 hyper-inflammatory syndrome. Numerous studies have described the relationship between systemic elevation of IL-6 and other diseases and systemic conditions, such as insulin resistance, type 1 and type 2 diabetes, diabetic nephropathy, and renal disease [2,12,13,14,15,16]. Moreover, TCZ has been studied for some time now in the field of kidney transplantation, firstly in antibody-mediated rejection (AMR). In 2017, Choi et al. published a study in which renal transplant patients with chronic active AMR and donor-specific antibodies were treated with TCZ after failing the standard of care treatment with intravenous immunoglobulins and rituximab (RTX) with or without plasma exchange [17]. The tocilizumab-treated patients demonstrated both high graft and patient survival rates at 6 years. Significant reductions in donor-specific antibodies and stabilization of renal function were seen at 2 years. This study has provided a valuable background for off-label TCZ use as a part of the treatment following kidney transplantation-AMR. In addition, TCZ use has been described as a part of the desensitization (DES) protocol, which allows kidney transplantation for highly human leukocyte antigen (HLA)-sensitized subjects. Based on the published studies, it seems that TCZ alone can reduce anti-HLA antibodies, but it is not sufficient as a monotherapy to allow transplantation; however, there is a possibility that combining TCZ with standard DES treatment could benefit the long-term outcome of HLA-incompatible transplants [18]. These findings were one of the main reasons why TCZ was chosen as a therapy option after csDMARD failure in our patient. It was considered reasonable to include TCZ in the therapy because there was a possibility that the treatment would benefit the post-transplantation course and would be included in the therapy anyhow. The main problem was its unknown safety regarding its use in patients on HD. In reviewing the literature, we found only two case reports that described safe and effective long-term administration of TCZ in HD patients [19,20]. In 2015, Mori et al. published the ACTRA-RI study, whose aim was to examine the efficacy of TCZ as a monotherapy versus the combination therapy of TCZ and MTX in patients with RA and renal insufficiency [21]. Their results show that there were no significant differences in efficacy between monotherapy and combination therapy, but also that the efficacy parameters in the renal insufficiency group were comparable with those in the group without this complication. Despite the positive outcomes considering TCZ monotherapy, there were only two patients with ESRD, of whom only one was on HD, from the total number of 102 patients with renal insufficiency included in the study. Naturally, general conclusions about the efficiency and safety of TCZ therapy in patients with ESRD, more precisely those on HD, cannot be drawn based on this one patient or the isolated cases mentioned previously, including ours, who did not experience adverse events regarding TCZ therapy. In the past few years, case reports have been published about TCZ therapy in patients with a COVID-19 infection and ESRD, but the drug has only been used short-term during the treatment and, of course, no conclusions about the safety of long-term administration of TCZ in HD patients in general can be drawn from these cases [12,22,23,24,25,26,27]. A treatment plan and decision should be guided by the patient’s individual needs and risks. In our case, the decision was carefully evaluated, all currently available therapeutic options were considered, and the Medical Affairs department of the manufacturer was consulted regarding the studies of TCZ in ESRD. While tumour necrosis factor inhibitors (TNFi) are usually the first choice of bDMARDs in the treatment of RA, in our case, the nephrologist favored TCZ. In our opinion, TNFi were the less preferred option since they are used as a combination therapy with csDMARDs. Considering the patient’s liver damage and the risk of infections, we preferred monotherapy, which, for TCZ, is proven as effective as a combination therapy. In addition, there are currently no official recommendations regarding the use of TNFi in patients with ESRD, although limited publications suggest its safe and efficient use [28,29]. One of the possible treatment options could be abatacept, a fusion protein that acts as a T-cell co-stimulation inhibitor, but since it is not registered in our country, its use was not thoroughly considered. Apart from that, its protective effect on the kidneys and proteinuria was described in some publications, and there are also papers supporting the use of abatacept instead of calcineurin inhibitors in post-transplant long-term immunosuppression [30,31]; however, as with TCZ, its elimination and behavior in ESRD remains understudied. In searching the published data, we did not come across a paper describing long-term therapy with abatacept in ESRD. Janus kinase (JAK) inhibitors were a novelty in our country on the RA treatment market when the decision regarding the patient’s therapy was made. Nevertheless, the results of recent studies show that patients taking JAK inhibitors are at higher risk of major adverse cardiovascular events, as well as malignancy and venous thromboembolism [32,33]. This resulted in updating of the product information for JAK inhibitors with new recommendations and warnings. Additionally, although it appears from the available data that renal impairment has a limited effect on JAK pharmacokinetics, we would not recommend JAK inhibitors treatment in HD patients generally [34,35]. HD patients have a major risk of cardiovascular incidents, which also represent the main mortality cause in this group of patients, making them ineligible for this group of drugs [36]. Rituximab, a monoclonal antibody directed against CD20, is not eliminated by hemodialysis, which makes it a good therapeutic option for patients with RA and ERDS, especially in those with the potential for a transplantation. It is included as a standard of care treatment for AMR. According to our Health Insurance Fund, RTX is approved as a third-line therapy, after unsuccessful treatment with csDMARD and TNFi, which is why it was not chosen as a first bDMARD option. All the above-mentioned drugs are associated with a higher risk of serious infection (SI). Considering her comorbidities, our patient was at even greater risk for a previously mentioned adverse event. A multi-database cohort study published in 2019 analyzed patients divided into two groups, the first being those who initiated TCZ or TNFi, and the second being those who initiated TCZ or abatacept [37]. All the included patients had a history of a prior use of ≥1 different biologic drugs or tofacitinib. The primary outcome was hospitalised SI. The results show no difference in the composite SI risk between TCZ and TNFi patients, but TCZ was associated with an increased risk of serious bacterial infection, skin and soft tissue infections, and diverticulitis. On the other hand, the risk of composite SI was higher in TCZ than abatacept patients; however, an observational cohort study conducted in Sweden and Denmark showed a lower incidence rate of SI in patients treated with TCZ compared to those receiving abatacept and RTX, with RTX having the highest incidence rate among the three. Apart from the drugs used in RA, the poorly regulated disease is also associated with an immunocompromised state due to dysregulation of the immune system [38]. In addition, a poorly regulated disease is also associated with several other risks, predominantly cardiovascular risk, and our patient has already developed coronary disease. Bearing in mind the risks and benefits, we considered that the chance of bringing the patient into remission and, thus, possible organ transplantation in the future, was of greater importance than the risk of infectious events, which, in our case, turned out to be the right decision because the patient has had no severe infections so far. If TCZ proves to be safe for administration in patients on HD, it could have more beneficial effects than other agents alone. Studies have shown that maintenance HD results in an inflammatory state, where between 30 and 50% of patients have elevated serum levels of inflammatory markers such as CRP and IL-6 [36,39,40,41,42]. The presence of inflammation is associated with an increased mortality risk, especially cardiovascular events, with IL-6 being the most robust predictor of comorbidity and outcomes in ESRD [36]. It seems that IL-6 plays a great role in the outcome for people on HD. In that manner, for RA patients with ESRD, TCZ as an IL-6 blocker could have more favorable results than the other therapeutic options, with the possibility of becoming the treatment of choice in the future, especially since a certain number of those patients will undergo renal transplantation.

## 4. Conclusions

The treatment of patients with RA and ESRD is a challenge for every rheumatologist and nephrologist. The number of rheumatologic medications that can be safely used in patients on HD is limited. Drugs needed in patients who do not respond to csDMARDs, both bDMARDs and targeted synthetic molecules, have not been systematically studied in ESRD patients and their use is mainly based on previous case reports in the literature. Based on our patient with RA on HD, we can conclude that TCZ could be the drug of choice in the treatment of patients with RA and ESRD, as it has been shown to be safe and effective over a 4-year period, but further studies are indeed needed. Furthermore, it is important to take an individualized approach when treating patients, keeping in mind that any drug can cause both serious adverse events and multiple beneficial effects, as TCZ has in our patient.

## Figures and Tables

**Table 1 medicina-59-01517-t001:** Laboratory and clinical data before and 4 years after the start of tocilizumab (TCZ) therapy.

	Reference Range	before TCZ	after 4 Years
White blood cell count (10^9^/L)	3.4–9.7	6.9	8.3
Neutrophils (%)	44–72	67.3	72
Lymphocytes (%)	20–46	20.7	20
Hemoglobin (g/L)	138–175	103	130
Platelets (10^9^/L)	158–424	343	391
CRP (mg/L)	<5.0	45.3	3
ESR (mm/3.6 ks)	2–20	33	18
Urea (mmol/L)	2.8–8.3	15.3	4.1
Creatinine (µmol/L)	49–90	722	221
eGFR CKD-EPI (mL/min/1.73 m^2^)	-	5	22
Aspartate aminotransferase (U/L)	8–30	17	21
Alanine aminotransferase (U/L)	10–36	16	34
Gamma-glutamyl transferase (U/L)	9–35	24	35
Alkaline phosphatase (U/L)	60–142	92	122
Total bilirubin (µmol/L)	3–20	7.7	10
Cholesterol (mmol/L)	<5.00	4.01	4.5
LDL cholesterol (mmol/L)	<3.00	3.05	2.9
HDL cholesterol (mmol/L)	1.20–5.00	0.99	1.2
Triglycerides (mmol/L)	<1.70	1.35	1.6
HbA1c (%)	<6.5	7.9	6.7
RF (IU/mL) *	<15.9	39.9	NA
Anti-CCP (EliA U/mL) *	Positive > 10	≥340	NA
Antinuclear antibody *	Positive > 1	0.1	NA
DAS28 CRP	High disease activity > 5.1	5.3	2.4

CRP: C-reactive protein, ESR: erythrocyte sedimentation rate, eGFR: estimated glomerular filtration rate, LDL: low-density lipoprotein, HDL: high-density lipoprotein, HbA1c: glycated hemoglobin (A1C, hemoglobin A1C, HbA1c), RF: rheumatoid factor, anti-CCP: anti-cyclic citrullinated peptide, DAS28: disease activity Score-28 CRP, NA: not applicable, * FEIA Phadia 200.

## Data Availability

The original data generated and analyzed for this study are included in the published article. Further inquiries can be directed to the corresponding author.

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
