# Peer review of "Safe and Efficient Use of Tocilizumab in Rheumatoid Arthritis Patient on Maintenance Hemodialysis: A Case Report"

_medicina, 2023, doi:10.3390/medicina59091517_

Round 1

Reviewer 1 Report

The authors describe a clinical case of rheumatoid arthritis associated with end-stage renal failure successfully treated with tocilizumab.

The work is interesting. however, it needs some clarifications:

1) Tocilizumab was administered weekly after hemodialysis session. However, the patient was undergoing hemodialysis three times a week. Could this have reduced the blood concentration of tocilizumab? Do the authors believe that more frequent administrations of tocilizumab may be more effective in patients on hemodialysis?
2) Treatment with tocilizumab is associated with increased infectious risk. Hemodialysis patients are generally immunocompromised. What is the infectious risk in treating such a patient with tocilizumab?
3) The patient was APCA positive. Could treatment with abatacept, which is less risky for infection, be an alternative? What about possible treatment with TNFi?
4) Some JAKi such as upadacitinib have no significant impact on renal function. Can they be in the authors' opinion an additional alternative to tocilizumab?

English needs only minor changes

Reviewer 2 Report

the authors must restructure the abstract ( maybe paraphrasing)

there are needed extensive english corrections

the subject is very important, kind of unmeet needs in dailly Rheumatology  practice 

I recommend to authors to realize a graphical image in the intoduction chapter with the glomerulofilration / hemodialysis 

I thought that the article is not suitable for  Life, but I verrified the literature and I was really surprised of the limited data on this topic ( more from COVID era- the authors cited these article) , I encourage the authors to restructure the article . 

more Discussion about all the antirheumatic drugs use  in hemodialysized patients 

-

Round 2

Reviewer 2 Report

the authors answerred according to Reviewers querries.